# Enhancement of loudness discrimination acuity for self-generated sound is independent of musical experience

**Nozomi Endo[1,2], Takayuki Ito[3,4], Katsumi Watanabe[5,6], Kimitaka Nakazawa[1]\***

**1** Department of Life Sciences, Graduate School of Arts and Sciences, The University of Tokyo, Tokyo, Japan, **2** Japan Society for the Promotion of Science, Tokyo, Japan, **3** CNRS, Grenoble INP, GIPSA-Lab, Univ. Grenoble Alpes, Grenoble, France, **4** Haskins Laboratories, New Haven, Connecticut, United States of America, **5** Faculty of Science and Engineering, Waseda University, Tokyo, Japan, **6** Faculty of Arts, Design and Architecture, University of New South Wales, Sydney, Australia

\* nakazawa@idaten.c.u-tokyo.ac.jp

**Data Availability Statement:** All relevant data are within the paper.

**Funding:** This study was jointly supported by research grants from JSPS KAKENHI (20J14743) and Tateishi Science and Technology Promotion

## Abstract

Musicians tend to have better auditory and motor performance than non-musicians because of their extensive musical experience. In a previous study, we established that loudness discrimination acuity is enhanced when sound is produced by a precise force generation task. In this study, we compared the enhancement effect between experienced pianists and non-musicians. Without the force generation task, loudness discrimination acuity was better in pianists than non-musicians in the condition. However, the force generation task enhanced loudness discrimination acuity similarly in both pianists and non-musicians. The reaction time was also reduced with the force control task, but only in the non-musician group. The results suggest that the enhancement of loudness discrimination acuity with the precise force generation task is independent of musical experience and is, therefore, a fundamental function in auditory-motor interaction.

## Introduction

Musicians have extensive auditory and motor experiences through their long-term training, and as a result, tend to have better auditory performance and corresponding motor performance than non-musicians. Psychoacoustic studies have shown that frequency discrimination thresholds are lower in musicians than in non-musicians, and the performance is dependent on the years of musical experience [1]. Pitch discrimination abilities are also better for musicians [2]. In addition to auditory ability, musicians have superior motor performance [3], superior perceptual acuity [4], and better somatosensory-motor interactions [5, 6]. Although these performances have been examined separately in auditory and motor functions, auditory and motor experiences via musical training may also affect the performance of auditory-motor interaction.

Previous studies have shown that auditory-motor interaction can alter auditory perception due to motor execution. The accuracy of auditory discrimination improved in the perception

Foundation Grant 2197001 awarded to N.E.; the National Institute on Deafness and Other Communication Disorders Grant R01-DC017439 awarded to T.I.; JSPS KAKENHI (17H00753 and 17H06344 to K.W.; 18H04082 to K.N.); JST Moonshot R&D (JPMJMS2012) to K.W. and K.N. The funders had no role in study design, data collection and analysis, decision to publish, or preparation of the manuscript.

**Competing interests:** The authors have declared that no competing interests exist.

of self-generated sounds [7] while this perception was attenuated [8]. We also found that explicit control of the produced force on a finger pressing for sound production (like a piano keystroke) improved loudness discrimination [9]. Auditory-motor interaction also biases the perception of sounds based on the context of movement or related somatosensory inputs. The pianist's perception of pitch is biased by the position of their keystroke on the piano [10]. The perception of speech sounds is biased depending on the somatosensory inputs associated with speech movement [11, 12]. These studies suggest that auditory-motor interactions can assist in the perception of self-generated sounds.

Although these findings on the perception of self-generated sound have been mainly investigated in online response or real-time perceptual processing, the auditory-motor interaction for the aid of auditory perception can be developed or improved by a specific experience, such as practicing the musical instrument. Considering that speech motor training changes the perception of speech sound [13, 14], musical experience and training could also improve auditory-motor interaction or representations for auditory perception, and musicians might show a different facilitatory effect compared to non-musicians.

The current study aimed to examine the effects of piano expertise on the change in loudness perception of sound produced by a precise force generation task. We focused on our previous finding that loudness discrimination acuity was facilitated when sound was self-generated with precise adjustment of force. We hypothesized that precision of motor execution, regardless of any amplitude of force, can be a key to improve the discrimination acuity. The precision of motor execution can be varied depending on the musical experience in a specific motor task such as the piano keystroke. We examined whether this facilitatory effect would depend on piano expertise and/or precise motor function with different force exertions by comparing pianists and non-musicians. If the piano expertise influences the perception of self-generated sounds through force-control task, a different amount of motor facilitation is expected. We also tested the two amplitudes of the target force. Although we found no change in the facilitatory effect depending on the amplitude of the produced force in non-musicians [9], pianists might show a change in facilitatory effect depending on the amplitude of the produced force due to their greater experience of sensorimotor performance on the piano playing.

## Materials and methods

### Participants

Thirty-four adults participated in the experiment (20–28 years). The participants were divided into two groups: the pianist group (n = 17) and the non-musician group (n = 17). The pianist group was assigned to those who majored in piano playing at a music college, and the participants of the pianist group had 13–24 years of piano experience. The non-musician group participants had not received such a professional musical education specifically concerning piano playing more than five years. Ten of them had never received a professional musical training. The participants were naïve to the purpose of the experiment. The experimental protocols were in accordance with the guidelines set out in the Declaration of Helsinki. All participants signed an informed consent form approved by the Waseda University Ethics Board (#2015–033).

### Settings

The main settings and procedures were the same as those in our previous study [9]. The participants were seated in front of a monitor (EV2450, EIZO) with headphones (HD280, Sennheiser). The right hand was used for the force generation task, and the left hand was used to respond to the auditory task by a keypress. In the force generation task, the force signals from

the sensor (USL06-H5-50N-D-FZ, Tec Gihan) were transferred to a laptop computer at a sampling frequency of 200 Hz via an analog-to-digital converter (NI 9215, National Instruments). Data acquisition and stimulus presentation were carried out using MATLAB (MathWorks, Inc.) with the Data Acquisition Toolbox (MathWorks, Inc.) and Psychophysics Toolbox extensions [15–17].

## Procedure

In the loudness discrimination test, two 1000-Hz pure tones (Fig 1A) were binaurally presented for 250 ms, separated by an interval of 1000 ms, through the headphones. The participants were asked to indicate whether the second sound (comparison stimulus) was louder or softer than the first sound (standard stimulus) by pressing the keys on the keyboard with their

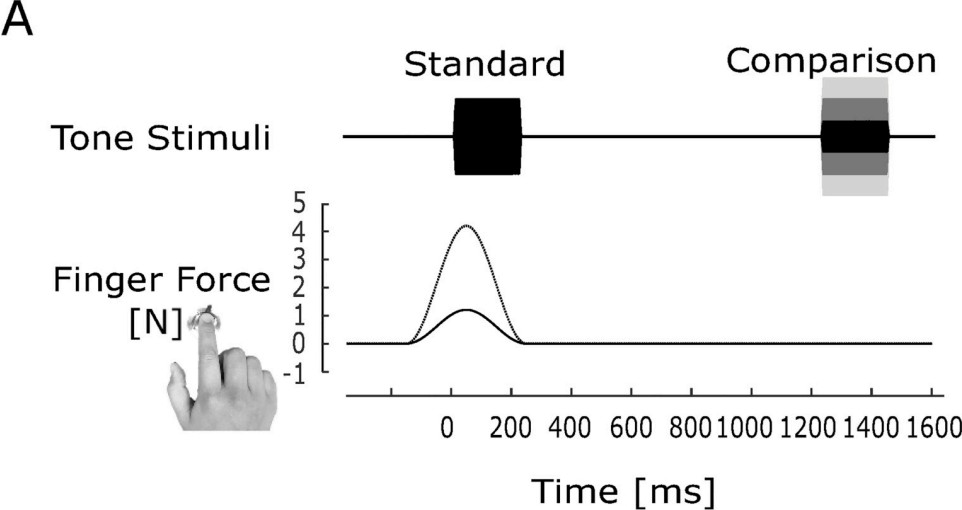

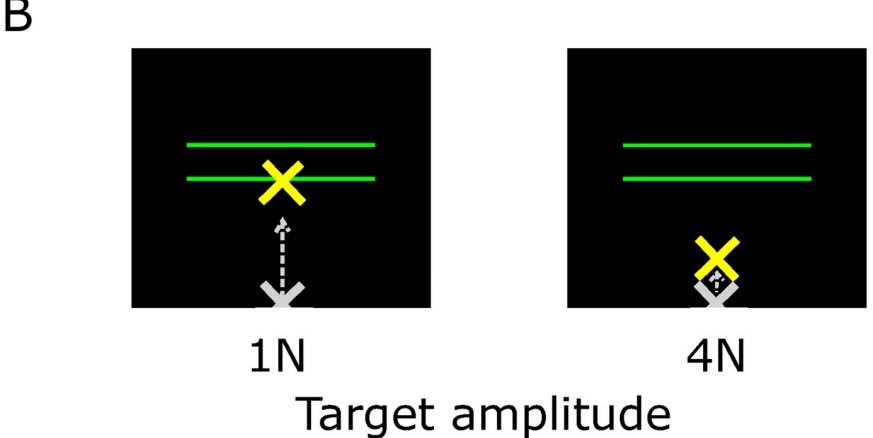

**Fig 1.** (A) Temporal patterns of auditory stimulation (top) and trajectory of finger force generation with two different target amplitudes (bottom). (B) Examples of visual presentation. The cross mark represents the cursor of the force amplitude. The two horizontal lines represent the target amplitude (bottom line) and upper limit (top line). The position of cursor represents the amplitude when a 1N force was generated. The gray cursor represents the start position of the cursor.

left hand. The amplitude of the standard stimulus was fixed at 65 dB, and the amplitude of the comparison stimulus was varied in each trial (from 62 dB to 68 dB in 1 dB increments). These amplitudes were determined as comfortable listening level based on participants reports in our preliminary experiment. Reaction times between the end of the comparison stimulus and participants' responses were recorded.

We tested two conditions: with the force generation task (motor-condition) and without the task (non-motor condition). In the motor condition, the force generation task was involved in presenting the standard stimulus, but not for the comparison stimulus. The participants were asked to produce a specific force amplitude by pressing the force sensor with their right index finger. They were instructed to complete the entire movement to reach the target amplitude and then release the force in a short period (approximately 500 ms). The amplitude of the produced force was presented on a monitor in real time as a vertical movement of a cross cursor mark (Fig 1B). The target and upper limits were also presented on the same monitor using a horizontal bar (Fig 1B). In order to make the same visual presentation, the upper limit was set 125% of target level. The sound stimulus was produced when the peak amplitude of the force was within the range of the target and the upper limit of the force. Participants practiced before the main experiment. When the force exceeded the upper limit, the trial was excluded and repeated in the subsequent trial. Based on our previous study, we tested two levels of the target force (1N and 4N) with an expectation of clear contrast between force conditions.

In the non-motor condition, participants placed their right hand gently on a desk without any force generation or movement. We replayed the force data recorded in the motor condition for visual presentation; the cursor moved, and the standard stimulus was played when the cursor reached the target level on a monitor, as in the motor condition. For this reason, the non-motor condition was performed after performing at least one session using the motor condition. We confirmed in a previous study that the order of the two tests did not interact with the change in discrimination acuity.

The experiment consisted of four sessions (two motor conditions and two non-motor conditions), with a 5-minute interval between sessions. In one session, 14 combinations (2 target force × 7 loudness levels) with ten repetitions were tested in random order (140 trials in total). We carried out two types of sequences for the sessions: 1) the motor and non-motor conditions were alternately conducted, or 2) the motor condition was repeated twice, and then the non-motor condition was repeated twice. The pattern of sessions was counterbalanced among participants.

## Data analysis

We calculated the probability that the participants perceived the second sound as louder for each amplitude of the comparison stimuli. A psychometric function was obtained by fitting cumulative Gaussians using a maximum-likelihood procedure separately for the motor and non-motor conditions. Based on the fitted psychometric curve, we obtained the point of subjective equality (PSE) and the just noticeable difference (JND). The JND was defined as half the difference of the comparison tone magnitude judged as louder on 75% and judged as louder on 25% of trials. Based on the previous study, we expected that the JND value would be smaller in the motor condition than in the non-motor condition, suggesting that the participant could better discriminate between the standard and comparison stimuli in the motor condition. The PSE was calculated at a 50% level of judgment probability in the estimated psychometric function. The higher PSE value represents that the participant perceived the standard stimulus as louder than the comparison stimulus. The JND, PSE, and reaction times were

analyzed using a mixed-design ANOVA. The between-participant variable was the experience factor (Pianist/Non-musician). Within-participant variables were motor factor (motor/non-motor) and force factor (1N/ 4N). For post-hoc tests, we applied a simple main effect test when the interaction was observed.

In order to examine a relationship between years of piano experience and the current enhancement effect, we examined a Pearson's product-moment correlation between number of years of piano experience and improvement of discrimination acuity. We also examined a correlation of years of piano experience with discrimination performance in non-motor and motor conditions respectively. In this analysis, averaged JNDs in the two force conditions were applied since we did not find any difference in force conditions (See Results). We carried out t-test to examine whether the correlations were significant.

## Results

Fig 2A shows the JND values for each motor condition across the two task force amplitudes. We found that the main effect of experience factor was significant ($F(1, 32) = 8.87$, $p < 0.01$, $\eta p^2 = 0.22$), indicating that the pianist group showed smaller JND values than the non-musician group. We also found that the main effect of motor factors was significant ($F(1, 32) = 15.00$, $p < 0.01$, $\eta p^2 = 0.32$), indicating that the motor condition led to smaller JND values than the non-motor condition. We did not find a significant main effect of force factor ($F(1, 32) = 2.88$, $p = 0.10$). No interactions were significant: experience × motor factors ($F(1, 32) = 1.24$, $p = 0.27$), motor × force factors ($F(1, 32) = 1.18$, $p = 0.29$), experience × force factors ($F(1, 32) = 0.00$, $p = 0.97$), and three-way interaction ($F(1, 32) = 0.21$, $p = 0.65$).

Additionally, we verified in detail a relationship between each condition by using a separate t-test in JND. We specifically focused on a comparison between the two groups in non-motor condition to verify a difference of basic performance, and a comparison between the motor condition of the non-musician group and the non-motor condition of musician group. Since we did not find any significant difference in the force factor, we took an average across force conditions. We found a reliable difference between the groups ($t(32) = 2.58$, $p < 0.05$), suggesting that the pianist group had a better auditory acuity than the non-musician group. In addition, JND in the motor condition of the non-musician group was not significantly different from that in the non-motor condition in the pianist group ($t(32) = 0.10$, $p = 0.93$), suggesting that the non-musician group improved the auditory acuity at the level of basic performance of the pianist group.

In analysis of Pearson's product-moment correlation, the number of years of piano experience was not significantly correlated with JND in non-motor condition ($r = 0.31$, $t(15) = 1.25$, $p = 0.23$), in motor condition ($r = 0.38$, $t(15) = 1.58$, $p = 0.13$), and JND difference between non-motor and motor condition ($r = 0.14$, $t(15) = 0.56$, $p = 0.58$).

In the PSE (Fig 2B), we found a significant difference in the force factor ($F(1, 32) = 13.40$, $p < 0.01$, $\eta p^2 = 0.30$). The PSE values in the 1N condition were significantly larger than those in the 4N condition. The other main effects were not significant: experience factor ($F(1, 32) = 1.17$, $p = 0.29$) or motor factor ($F(1, 32) = 0.36$, $p = 0.55$). We also did not find any interaction effect: experience × motor factors ($F(1, 32) = 0.63$, $p = 0.43$), experience × force factors ($F(1, 32) = 0.36$, $p = 0.55$), and motor × force factors ($F(1, 32) = 0.01$, $p = 0.94$), or thee-way interaction ($F(1, 32) = 0.43$, $p = 0.52$).

As for the reaction times, we found a significant interaction between experience and motor factors ($F(1, 32) = 5.56$, $p < 0.05$; $\eta p^2 = 0.15$). We did not find any significant difference in any of the three main effects: experience factor ($F(1, 32) = 0.27$, $p = 0.61$), motor factor ($F(1, 32) = 2.67$, $p = 0.11$), and force factor ($F(1, 32) = 0.61$, $p = 0.44$)], or in the other interaction effects:

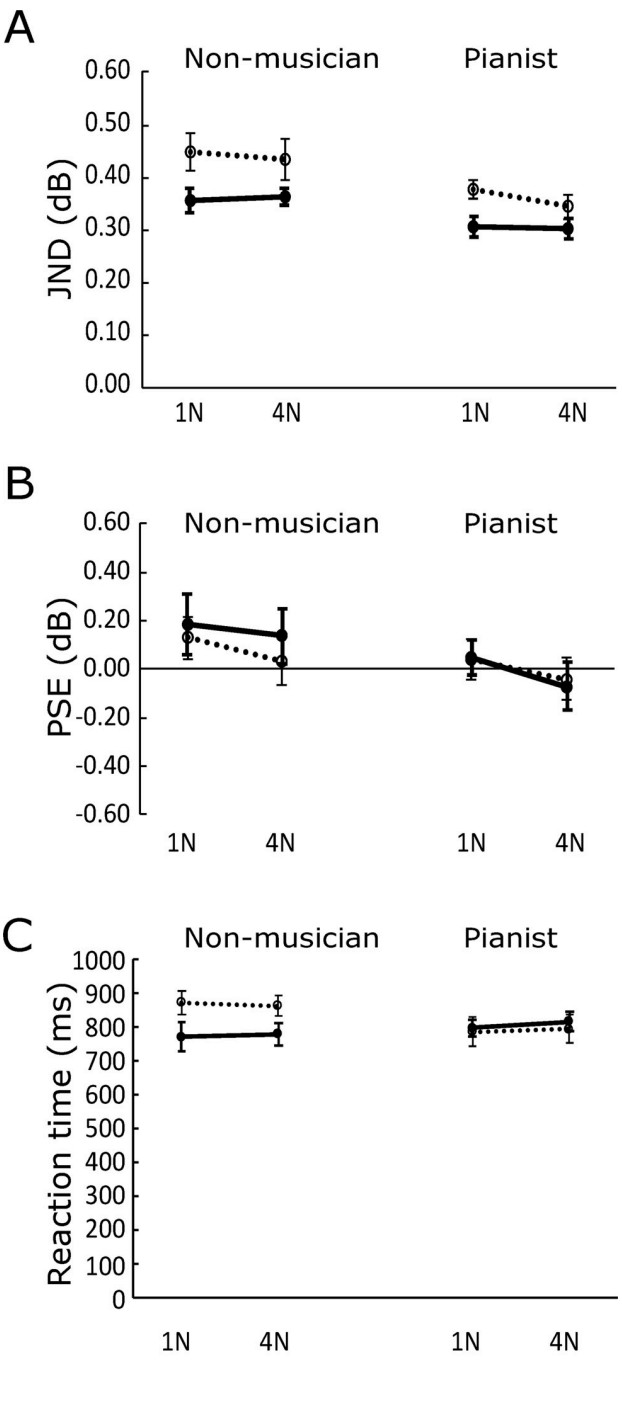

**Fig 2.** Mean values of (A) just-noticeable difference (JND), (B) point of subjective equality (PSE) and (C) reaction time. The solid line with filled circles represents motor condition, and the dashed line with open circles represents non-motor condition. Error bars show the standard error across the participants. PSE is represented as a value relative to 65 dB, whichis the amplitude of standard stimulus.

piano × force factors ($F(1, 32) = 0.94$, $p = 0.34$), motor × force factors ($F(1, 32) = 1.04$, $p = 0.32$), and three-way interaction ($F(1, 32) = 0.06$, $p = 0.81$). Simple main effect tests showed that the difference in reaction time between the motor and non-motor conditions was significant in the non-musician group ($F(1, 32) = 8.00$, $p < 0.01$; $\eta p^2 = 0.31$), but not in the pianist group ($F(1, 32) = 0.26$, $p = 0.62$). On the other hand, simple main effects between groups were not significant in the motor condition ($F(1, 32) = 0.42$, $p = 0.52$) or in the non-motor condition ($F(1, 32) = 2.46$, $p = 0.12$). The results suggested that the motor task differently affected the processing time for the loudness discrimination task depending on the musical experience.

In summary, we found that discrimination of sound loudness improved when the sound was generated by the motor task in both the pianist and non-musician groups. In basic auditory performance (without motor task), the pianist group showed better discrimination acuity than the non-musician group. In perceptual bias, the presented sound in the 1N condition was perceived louder than in the 4N condition in both motor and non-motor conditions. The non-musician group reacted faster in the motor condition than in the non-motor condition. However, we did not find such a difference in the pianist group.

## Discussion

The current study aimed to examine whether the change in loudness discrimination acuity for self-generated sounds [9] was dependent on musical experience. We compared the changes in loudness discrimination acuity between pianist and non-musician groups. The pianist group showed generally higher discrimination acuity than the non-musician group, consistent with pianists having better auditory perception than non-musicians because of their musical training [1, 2, 18]. Since there were not significant correlations between the number of years of piano experience and discrimination acuity, having education at a music college, regardless of the number of years, may be important. As shown in a previous study, the motor task to generate stimulus sounds improved the loudness discrimination acuity. The magnitudes of change were similar in both groups, as indicated by the lack of a statistically significant interaction. The number of years of piano experience in the pianist group was not correlated with the amount of enhancement of discrimination acuity. Contrary to our expectation that pianists would be more sensitive to different amplitudes of the produced force for self-generated sound, we found no modulation related to the amplitude of the produced force in either group. As a separate effect resulting from the motor task, we found that reaction time was facilitated in the non-musician group but not in the pianist group. In summary, the current results indicate that sound generation movement enhances loudness discrimination acuity similarly in the pianist and non-musician groups.

As seen in musical experts, there is the idea that auditory-motor experience via long-time training results in better auditory abilities [19, 20]. Improvements in auditory ability were seen in non-musicians with a brief training period as a change in auditory cortical responses [21, 22]. However, it is unclear what components in auditory-motor training help to improve auditory performance, although auditory-motor interactions are expected to contribute. The present study showed that the auditory-motor task itself could improve auditory performance in online processing.

Musical experience and training can improve somatosensory-motor interactions or representations [5, 6]. As a result, musical experts showed better precision in somatosensory processing and motor performance. We expected that better performance in somatosensory and motor processing might also affect the processing of auditory sounds produced by their own movements. However, experience-dependent changes in auditory-motor interactions were not observed in the current context. Although we did not use a piano keyboard for our motor

task to investigate somatosensory and motor precision as in other studies, the possible effect of using a device other than a piano keyboard for motor tasks would be small because the effect of pitch perception due to the pianist's finger movements was similarly induced in both conditions—with a piano keyboard and a computer keyboard [10]. Since we found a similar change in discrimination acuity due to motor tasks in the pianist and non-musician groups, the auditory-motor interaction may not be the function acquired or improved through training or development; rather, it can be a more basic process in auditory-motor function.

The results showed that the motor task facilitated the reaction times of loudness discrimination in the non-musician group, but not in the pianist group. In general, reaction time reflects processing time concerning sound encoding, decision making by comparing the sounds, and motor execution to respond. This processing time can be affected by sensorimotor experience and cognitive load. In our result, the reaction time of the non-musician group in the motor condition was similar to that of the pianist group. This difference between pianists and non-musicians indicates a difference in the processing time of the auditory task. Considering that pianists respond to auditory presentation faster than non-musicians due to musical experience [23], the pianist group may be familiar with comparing sound loudness and discriminating loudness differences, and thereby, showed similar performance in both motor and non-motor conditions such as flooring effect. Meanwhile, non-musicians may require more time for loudness discrimination in externally generated sounds. Since the motor task improved reaction time in the non-musician group to raise them to the same level of basic performance as the pianist group, the motor task can also help non-musicians have the same level of ability as a musician in loudness discrimination. This may be because sound loudness can be perceived more correctly with the motor task in the auditory processing prior to the judgment of loudness discrimination.

In our results, the interaction effect in reaction time was found, while it was not seen in the JND results. Our previous study [9] showed that the change in reaction time did not always correlate with the change in JND, suggesting the possibility of the modulation of reaction time arising from a source other than the one for the current facilitatory effect in loudness discrimination. Considering a debate between whether the perceptual changes in sound generation movement are influenced by the motor function itself and whether they are associated with changes in higher cognitive functions [24–26], the current perceptual changes in loudness discrimination acuity may be ascribed to the former. This possible interaction mechanism can also support the idea that the current facilitatory effect is a fundamental function that is not influenced by musical experience.

We found that the PSE values changed between force conditions. Since this change was seen in both motor and non-motor conditions, this can be due to the effect of visual presentation. A larger contrast of visual cursor speed, which is related to a larger contrast of the force, may induce changes in PSE value of auditory perception. This is consistent with previous studies, including our previous study, a contrast of visual information affects auditory perception [27–30]. Although this visual influence on auditory perception is important to understand sensory mechanism, we did not pursue this effect in the current study since this is beyond our scope.

Our results also showed that the discrimination acuity in the non-musician group was improved to the same level as the base performance in the pianist group (passive listening without sound generation movement). This suggests that non-musicians can possibly perform at a similar level as musicians' auditory perception when a sound is produced by a specific type of motor task, such as practicing with a musical instrument. Considering the experience-independent nature of the current facilitatory effect, a repetitive experience of better auditory situations with motor tasks may play a role in acquiring better auditory performance as a result of training with a musical instrument.

Temporal consistency is important for the interaction between auditory processing and other modality of processing including motor execution [31–35]. In the study of self-generated sound, delay of 200 ms or more are particularly critical to induce auditory cortical responses of self-generated sound [33]. In the current study, we tested synchronous condition that the auditory stimulus was produced when the motor task was achieved. This 200 ms of time window may be also applied into the current enhancement of discrimination acuity. Further investigation is required.

In conclusion, the current study demonstrated that both non-musicians and pianists similarly benefited from the enhancement of sound discrimination with the auditory-motor task. The study also reaffirmed that musicians have greater auditory ability than non-musicians. The current study has strengthened the understanding of the perception of self-generated sound and increased our knowledge about how perception can be more accurate using motor function [7, 9]. These findings may have implications for auditory training through playing musical instruments and the mechanism of musical expertise.

## Author Contributions

**Conceptualization:** Nozomi Endo, Takayuki Ito, Katsumi Watanabe, Kimitaka Nakazawa.

**Data curation:** Nozomi Endo.

**Formal analysis:** Nozomi Endo.

**Funding acquisition:** Nozomi Endo, Takayuki Ito, Katsumi Watanabe, Kimitaka Nakazawa.

**Investigation:** Nozomi Endo, Takayuki Ito.

**Supervision:** Takayuki Ito, Katsumi Watanabe, Kimitaka Nakazawa.

**Writing – original draft:** Nozomi Endo, Takayuki Ito, Katsumi Watanabe, Kimitaka Nakazawa.

**Writing – review & editing:** Takayuki Ito, Katsumi Watanabe, Kimitaka Nakazawa.

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
