## [Decision Letter · Decision Letter 0]

24 Sep 2021

PONE-D-21-23155Enhancement of loudness discrimination acuity for self-generated sound is independent of musical experiencePLOS ONE

Dear Dr. Nakazawa,

Thank you for submitting your manuscript to PLOS ONE. After careful consideration, we feel that it has merit but does not fully meet PLOS ONE’s publication criteria as it currently stands. Therefore, we invite you to submit a revised version of the manuscript that addresses the points raised during the review process.

I should apologize for taking so much time to response to the authors at this moment. I received this manuscript at the end of July, and from there I sought the reviewers, and the reviewers responded promptly. But as a result, it was a so late response from the editor to the authors.The comment from the reviewers have all been positive and I believe that a few corrections would greatly improve the manuscript. 

We look forward to receiving your revised manuscript.

Kind regards,

Kenichi Shibuya, Ph.D.

Academic Editor

PLOS ONE

Journal Requirements:

Reviewers' comments:

Reviewer's Responses to Questions

**Comments to the Author**

1. Is the manuscript technically sound, and do the data support the conclusions?

Reviewer #1: Yes

Reviewer #2: Yes

Reviewer #3: Yes

2. Has the statistical analysis been performed appropriately and rigorously? 

Reviewer #1: Yes

Reviewer #2: Yes

Reviewer #3: Yes

3. Have the authors made all data underlying the findings in their manuscript fully available?

Reviewer #1: Yes

Reviewer #2: No

Reviewer #3: Yes

4. Is the manuscript presented in an intelligible fashion and written in standard English?

Reviewer #1: Yes

Reviewer #2: Yes

Reviewer #3: Yes

5. Review Comments to the Author

Reviewer #1: In the submitted manuscript, the authors conducted a study to examine whether loudness discrimination acuity is more enhanced by musical experience.

34 young participants (17 pianists) listened to two sequential pure tones and were asked to sort out perceived loudness. In motor task condition, the first tone was produced when a load sensor was pressed by a finger (with a force of 1N or 5N). Several performance metrics such as just noiceable difference, point of subjective equality, and reaction time were measured.

Overally, the manuscript was well organized and the results were reasonably communicated. This paper is almost ready to be accepted by Plos one journal, however, the manuscript has several weaknesses that should be addressed prior to publication:

1. Line 80. The authors recruited non-musical participants, who have less than five years of musical experience. However, the term musical experience seems somewhat vague and should be further specified.

2. Line 97. Did the authors measure audiograms? Since this study used preset conditions of sound pressure (i.e., 62~68dB SPL), individual difference of hearing ability might have affected the results.

3. Line 102. Please provide a short reason why 65 dB SPL (why not 70~80?) was set as standard stimulus.

4. Line 125. Even if the previous work of the authors used an upper limit of the production force as target level + 0.5 N, the value should be specified in this manuscript as well for reader's comprehension. Furthermore, authors would be better to explain why the fixed value (0.5 N) is used rather than relative value (e.g., 120% of target level).

5. Line 126. The present study used force conditions of 1 N and 4 N. However, why these values were selected needs to be further addressed.

6. Line 175. Please provide more information on how to interpret the plot, specifically the PSE. The higher value means the perceived loudness is overrated?

7. Line 267. Please specify the meaning of reaction time in this study. One might not understand that it reflects composite stages of cognitive processing such as encoding and mental processing.

8. Line 277. This is a major concern. A clarification and deeper explanation is dutiful on the effects of force conditions obtained in this study. How visual stimuli affected auditory perception? Is it the sole cause of the current results?

Reviewer #2: The current study examined whether musical experience modulates an effect of loudness discrimination improvement by a self-generated motor response. Specifically, following the author’s previous study (Endo et al., 2021) that precise self-generated action improves loudness discrimination, in the current study, they further examined whether the previously found motor-auditory enhancement differs between musicians (pianists) and non-musician groups. Results showed no significant difference between the groups in the motor-auditory enhancement, whereas pianists generally showed better auditory acuity than non-musicians.

The motor-auditory enhancement phenomenon is interesting, the research motivation is clear, and the manuscript is well written. However, I have some concerns about the null result and its interpretations. Details are listed in the following.

First of all, it is unclear why the self-generated action improves loudness discrimination. Without a clear hypothesis, it is confusing whether enhanced motor-sound training experience like pianists would show stronger or weaker motor-auditory enhancement effect. It made it difficult to judge whether the non-different motor-auditory enhancement between the groups is due to other factors like floor effect or some kind of cancelation among different factors. If the authors try more factors/parameters, may they find the modulation by the musical experience?

Related to the first issue, in the abstract, the authors state that “the loudness discrimination acuity is enhanced when sound is produced by a precise force generation task,” what does “precise” mean? Does it refer to the timing of the sound and the action, or other correspondence between them? In other words, is the synchrony between the sound and action critical to evoke this motor-auditory enhancement? Or does the relationship between the action force and sound intensity modulate the effect? I am not sure if the authors have explored these factors in their previous studies, but if yes, it would be clearer to elaborate more about the possible underlying mechanism of this motor-auditory enhancement phenomenon and to make a clearer hypothesis as to how enhanced motor-sound training influences the effect. As the authors mentioned in p.11, “pianist might show a change in facilitatory effect depending on the amplitude of the produced force due to their greater experience of sensorimotor performance on the piano playing,” is it possible that the result would be different if more parameters are investigated, such as the synchrony between the sound and action and the correspondence of the finger force and sound intensity?

In Abstract, line 24, “Without the force generation task, loudness discrimination acuity and reaction time were better in pianists than non-musicians in the condition,” the statement about the reaction time seems incorrect as “the simple main effects between groups were not significant in the motor condition or in the non-motor condition” (p.19, line 201-204).

In the Result session, the second paragraph, line 177-185, I am not sure if it makes sense to perform the t-tests there, under the condition the ANOVA results have been reported.

In the Discussion session, line 236, “The present study showed that the auditory-motor task itself could improve auditory performance in online processing.” Can the task be considered an auditory-motor task, while the participant did not need any coordination between the sound and motor demand (if my understanding is correct)?

Reviewer #3: This manuscript examined the effect of musical experience on enhancement of loudness discrimination acuity in conjunction with a force generation task. This is of interest, but this reviewer raises some concerns.

MAJOR

It needs to be argued whether it is reasonable to use the enhancement of loudness discrimination acuity to evaluate the pianist's auditory-motor interaction with the force generation task that took place one second earlier. One second seems like a considerable amount of time in the flow of the music, so it is doubtful that the work done one second later has anything to do with the music.

The authors are examining the influence of musical experience, so they need to describe the influence of years of experience or level of music.

If the effects cannot be described, it is not possible to discuss the effects of music experience, though negative. Because the effects could be due to acquirement by musical practice and innate effects. Besides this is a cross-sectional study. A lot of wording needs to be changed to cautious expressions in this regard, e.g., L67 piano experience.

MINOR

L 55. Aide (typo)

6. PLOS authors have the option to publish the peer review history of their article (what does this mean?). If published, this will include your full peer review and any attached files.

Reviewer #1: No

Reviewer #2: No

Reviewer #3: No

---

## [Author Response · Author response to Decision Letter 0]

12 Oct 2021

Response to Reviewer 1:

1. Line 80. The authors recruited non-musical participants, who have less than five years of musical experience. However, the term musical experience seems somewhat vague and should be further specified.

-We specified this point at ll.83-85.

2. Line 97. Did the authors measure audiograms? Since this study used preset conditions of sound pressure (i.e., 62~68dB SPL), individual difference of hearing ability might have affected the results.

-We did not measure audiograms. We think individual difference in hearing threshold would not be too much of issue in our context, since we focused on a relative change between conditions. We expect that the change can be induced regardless of any difference in hearing threshold. In addition, since the sound pressure level that we used was enough above the hearing threshold that can be expected in the participants who have normal hearing, the participants can be perceived our stimulus sound effortlessly. This is supported by the measured JND value in the current task that was smaller than 1dB. 

3. Line 102. Please provide a short reason why 65 dB SPL (why not 70~80?) was set as standard stimulus.

-We set this amplitude as comfortable level by participants reports in our preliminary experiment. We added this explanation at ll.109-110.

4. Line 125. Even if the previous work of the authors used an upper limit of the production force as target level + 0.5 N, the value should be specified in this manuscript as well for reader's comprehension. Furthermore, authors would be better to explain why the fixed value (0.5 N) is used rather than relative value (e.g., 120% of target level).

-In the previous study (Endo et al., 2021), the target level + 0.5N for the upper limit was used in the Experiment1 when we present a target level was presented in a different height. The procedure in the current paper was similar to Experiment2 when we present a target level at the same height with the different velocity of cursor movement. In order to make the same visual presentation across different force conditions, we used a relative value, which was 125% of target level for the upper limit. As showed in the previous study, the difference concerning the upper limitation does not affect the current effect of JND change. We clarified the setting of upper limitation at ll.128-129.

5. Line 126. The present study used force conditions of 1 N and 4 N. However, why these values were selected needs to be further addressed.

-Based on our preliminary experiment, 1N is a minimum level to achieve the current task with our experimental setup, and 4N is a maximum level that the participants are able to complete the current experiment without feeling much of fatigue. With an expectation of a clear contrast between force conditions, we selected these two force conditions based on our previous study. We added this point at ll.133-134.

6. Line 175. Please provide more information on how to interpret the plot, specifically the PSE. The higher value means the perceived loudness is overrated?

-We added an explanation at ll.162-163.

7. Line 267. Please specify the meaning of reaction time in this study. One might not understand that it reflects composite stages of cognitive processing such as encoding and mental processing.

-We added an explanation at ll.281-284 in the previous paragraph instead, since reaction time was first shown up there.

8. Line 277. This is a major concern. A clarification and deeper explanation is dutiful on the effects of force conditions obtained in this study. How visual stimuli affected auditory perception? Is it the sole cause of the current results?

-Considering our experimental procedure, we think the visual presentation can be a possible source to induce changes in PSE value. As shown in previous studies, a contrast of visual stimuli affects auditory perception. Although we fully agreed that this visual effect in auditory perception is an interesting topic, this is beyond the scope of the present paper and we would like to pursue this topic in the future. We clarified this point at ll.309-313. 

 

Response to Reviewer 2:

1.First of all, it is unclear why the self-generated action improves loudness discrimination. Without a clear hypothesis, it is confusing whether enhanced motor-sound training experience like pianists would show stronger or weaker motor-auditory enhancement effect. It made it difficult to judge whether the non-different motor-auditory enhancement between the groups is due to other factors like floor effect or some kind of cancelation among different factors. If the authors try more factors/parameters, may they find the modulation by the musical experience?

-We clarified our hypothesis in the Introduction (ll.64-67). In the current experiment, we found a different baseline depending on the musical experience, since the JND was different between pianists and non-musicians in the non-motor condition. For the enhancement with motor task, which is the focus of this study, there was no difference between pianists and non-musicians. As an additional analysis, we tried to correlate enhancement with the number of years of experience within the pianist, but there was no significant correlation (see ll.203-210). Therefore, we conclude that motor enhancement is independent of experience.

2.Related to the first issue, in the abstract, the authors state that “the loudness discrimination acuity is enhanced when sound is produced by a precise force generation task,” what does “precise” mean? Does it refer to the timing of the sound and the action, or other correspondence between them? In other words, is the synchrony between the sound and action critical to evoke this motor-auditory enhancement? Or does the relationship between the action force and sound intensity modulate the effect? I am not sure if the authors have explored these factors in their previous studies, but if yes, it would be clearer to elaborate more about the possible underlying mechanism of this motor-auditory enhancement phenomenon and to make a clearer hypothesis as to how enhanced motor-sound training influences the effect. 

-As for the word "precise", we use it in the meaning that the force is generated close to the target with precise adjustment. In the sound generation task used in previous studies, this precision of force generation has not been taken into account because the previous demands of motor task were to overshoot the produced force by a specific threshold for sound generation, in which the amplitude of force was depended on the individuals. As a result, the results in those previous studies were not consistent, such as enhancement or attenuation. Therefore, “precise force control”, which is used in our studies, is the key to enhancement of discrimination acuity. This point was clarified in the Introduction by responding the first question. 

 Regarding the timing, we did not test in the current study. In response to a comment of Reviewer 3, we added one paragraph about the topic for timing in the Discussion (ll.322-332).

 As for the difference in the magnitude of the force, we reported in the current paper that there is no effect into the current auditory task. This result is consistent with our previous results (Endo et al. 2021). 

3. As the authors mentioned in p.11, “pianist might show a change in facilitatory effect depending on the amplitude of the produced force due to their greater experience of sensorimotor performance on the piano playing,” is it possible that the result would be different if more parameters are investigated, such as the synchrony between the sound and action and the correspondence of the finger force and sound intensity?

-We added a correlation analysis between the year of piano experience and amplitudes of the current effect. The result still supports the current findings that the enhancement of discrimination acuity is not related to musical experience. In the current context, we conclude that the current effect can be independent of musical experience and it could be possible to find the dependent effect in a different context. 

4. In Abstract, line 24, “Without the force generation task, loudness discrimination acuity and reaction time were better in pianists than non-musicians in the condition,” the statement about the reaction time seems incorrect as “the simple main effects between groups were not significant in the motor condition or in the non-motor condition” (p.19, line 201-204).

-Reviewer is right. We corrected the sentence (l.24). Thanks.

5. In the Result session, the second paragraph, lines 177-185, I am not sure if it makes sense to perform the t-tests there, under the condition the ANOVA results have been reported.

-We applied t-test as a post-hoc analysis to verify this enhancement in detail between each condition. Specifically, we would like to verify a difference in basic auditory performance (non-motor condition) between the groups, and to verify how much performance in non-musician group was improved due to motor task by refering base auditory performance in musician group. We clarified this point in ll. 198-202.

6. In the Discussion session, line 236, “The present study showed that the auditory-motor task itself could improve auditory performance in online processing.” Can the task be considered an auditory-motor task, while the participant did not need any coordination between the sound and motor demand (if my understanding is correct)?

-The current task can be considered as auditory-motor task because the stimulus sound was presented with achievement of the motor task that is to reach the produced force at a target force amplitude. Specifically, by setting an upper limit, the participants were required to produce precisely the target force amplitude. We described this point in the Methods. 

 

Response to Reviewer 3:

 MAJOR

1. It needs to be argued whether it is reasonable to use the enhancement of loudness discrimination acuity to evaluate the pianist's auditory-motor interaction with the force generation task that took place one second earlier. One second seems like a considerable amount of time in the flow of the music, so it is doubtful that the work done one second later has anything to do with the music.

-We added a paragraph about a possibility of temporal synchrony in the Discussion (ll.322-332).

2. The authors are examining the influence of musical experience, so they need to describe the influence of years of experience or level of music.

 If the effects cannot be described, it is not possible to discuss the effects of music experience, though negative. Because the effects could be due to acquirement by musical practice and innate effects. Besides this is a cross-sectional study. A lot of wording needs to be changed to cautious expressions in this regard, e.g., L67 piano experience.

-We added a correlation analysis between years of musical experience and amplitude of the current enhancement effect. The result support the idea that musical experience is independent of the current enhancement effect. We added this result in the Results and Discussion sections. As a response to Reviewer1's comment, we also added an explanation concerning the level of music for the non-musician group.

 MINOR

3. L 55. Aide (typo)

-We corrected. Thanks.

---

## [Decision Letter · Decision Letter 1]

11 Nov 2021

PONE-D-21-23155R1Enhancement of loudness discrimination acuity for self-generated sound is independent of musical experiencePLOS ONE

Dear Dr. Nakazawa,

Thank you for submitting your manuscript to PLOS ONE. Only Reviewer 2 has asked the authors to make one correction. Please respond to the comment raised by Reviewer 2 and re-submit the revised manuscript. 

We look forward to receiving your revised manuscript.

Kind regards,

Kenichi Shibuya, Ph.D.

Academic Editor

PLOS ONE

Journal Requirements:

Reviewers' comments:

Reviewer's Responses to Questions

**Comments to the Author**

1. If the authors have adequately addressed your comments raised in a previous round of review and you feel that this manuscript is now acceptable for publication, you may indicate that here to bypass the “Comments to the Author” section, enter your conflict of interest statement in the “Confidential to Editor” section, and submit your "Accept" recommendation.

Reviewer #1: All comments have been addressed

Reviewer #2: All comments have been addressed

Reviewer #3: All comments have been addressed

2. Is the manuscript technically sound, and do the data support the conclusions?

Reviewer #1: Yes

Reviewer #2: Yes

Reviewer #3: Yes

3. Has the statistical analysis been performed appropriately and rigorously? 

Reviewer #1: Yes

Reviewer #2: Yes

Reviewer #3: Yes

4. Have the authors made all data underlying the findings in their manuscript fully available?

Reviewer #1: No

Reviewer #2: Yes

Reviewer #3: Yes

5. Is the manuscript presented in an intelligible fashion and written in standard English?

Reviewer #1: Yes

Reviewer #2: Yes

Reviewer #3: Yes

6. Review Comments to the Author

Reviewer #1: (No Response)

Reviewer #2: The revision clarified the preciously raised concerns. I appreciate the author’s efforts and recommend publication. There is just one minor comment about the added correlation analyses. It was not described clearly how exactly the author performed the analyses. For example, did they perform Pearson or Spearman correlation, or others? Did they perform the t-test on the correlations (l. 210-214)?

Reviewer #3: (No Response)

7. PLOS authors have the option to publish the peer review history of their article (what does this mean?). If published, this will include your full peer review and any attached files.

Reviewer #1: No

Reviewer #2: **Yes: **Hsin-I Liao

Reviewer #3: No

---

## [Author Response · Author response to Decision Letter 1]

14 Nov 2021

Response to Reviewer #2:

Comment: The revision clarified the preciously raised concerns. I appreciate the author’s efforts and recommend publication. There is just one minor comment about the added correlation analyses. It was not described clearly how exactly the author performed the analyses. For example, did they perform Pearson or Spearman correlation, or others? Did they perform the t-test on the correlations (l. 210-214)?

Response: We carried out Pearson’s product-moment correlation (Pearson’s correlation) and t-test on the correlations. We added this point in the method (l.170, 175) and the result (l.207). Thanks.

---

## [Editor Report · Decision Letter 2]

18 Nov 2021

Enhancement of loudness discrimination acuity for self-generated sound is independent of musical experience

PONE-D-21-23155R2

Dear Dr. Nakazawa,

We’re pleased to inform you that your manuscript has been judged scientifically suitable for publication and will be formally accepted for publication once it meets all outstanding technical requirements.

Kind regards,

Kenichi Shibuya, Ph.D.

Academic Editor

PLOS ONE
---

## [Editor Report · Acceptance letter]

25 Nov 2021

PONE-D-21-23155R2 

Enhancement of loudness discrimination acuity for self-generated sound is independent of musical experience 

Dear Dr. Nakazawa:

I'm pleased to inform you that your manuscript has been deemed suitable for publication in PLOS ONE. Congratulations! Your manuscript is now with our production department. 

Kind regards, 

on behalf of

Dr. Kenichi Shibuya 

Academic Editor

PLOS ONE